# Is "no test is better than a bad test"? Impact of diagnostic uncertainty in mass testing on the spread of COVID-19

**Nicholas Gray**[1][☉]*, **Dominic Calleja**[1][☉], **Alexander Wimbush**[1][☉], **Enrique Miralles-Dolz**[1], **Ander Gray**[1], **Marco De Angelis**[1], **Elfriede Derrer-Merk**[1], **Bright Uchenna Oparaji**[1], **Vladimir Stepanov**[1], **Louis Clearkin**[2], **Scott Ferson**[1]

**1** Institute for Risk and Uncertainty, University of Liverpool, Liverpool, United Kingdom, **2** Wirral & Liverpool University Teaching Hospitals, Birkenhead, United Kingdom

☉ These authors contributed equally to this work.
* nickgray@liverpool.ac.uk, COVID19@riskinstitute.uk

**Data Availability Statement:** https://github.com/Institute-for-Risk-and-Uncertainty/SIRQ-imperfect-testing.

## Abstract

Testing is viewed as a critical aspect of any strategy to tackle epidemics. Much of the dialogue around testing has concentrated on how countries can scale up capacity, but the uncertainty in testing has not received nearly as much attention beyond asking if a test is accurate enough to be used. Even for highly accurate tests, false positives and false negatives will accumulate as mass testing strategies are employed under pressure, and these misdiagnoses could have major implications on the ability of governments to suppress the virus. The present analysis uses a modified SIR model to understand the implication and magnitude of misdiagnosis in the context of ending lockdown measures. The results indicate that increased testing capacity alone will not provide a solution to lockdown measures. The progression of the epidemic and peak infections is shown to depend heavily on test characteristics, test targeting, and prevalence of the infection. Antibody based immunity passports are rejected as a solution to ending lockdown, as they can put the population at risk if poorly targeted. Similarly, mass screening for active viral infection may only be beneficial if it can be sufficiently well targeted, otherwise reliance on this approach for protection of the population can again put them at risk. A well targeted active viral test combined with a slow release rate is a viable strategy for continuous suppression of the virus.

## Introduction

During the early stages of the United Kingdoms SARS-CoV-2 epidemic, the British government's COVID-19 epidemic management strategy was been influenced by epidemiological modelling conducted by a number of research groups [1, 2]. The analysis of the relative impact of different mitigation and suppression strategies concluded that the "only viable strategy at the current time" is to suppress the epidemic with all available measures, including the lockdown of the population with schools closed [3, 4]. Similar analysis in other countries lead to over half the world population being in some form of lockdown by April 2020 and over 90% of

**Funding:** This work has been partially funded through the following grants: UK Engineering and Physical Science Research Council (EPSRC) IAA exploration award with grant number EP/R511729/1 (NG, AW, SF, MDA) EPSRC and Economic and Social Research Council (ESRC) Centre for Doctoral Training in Quantification and Management of Risk and Uncertainty in Complex Systems and Environments, EP/L015927/1 (AW, DC, EMD, AG, VS, EDM) UK Medical Research Council (MRC) "Treatment According to Response in Giant cEll arTeritis (TARGET)", MR/N011775/1 (BUO, LC, SF) EPSRC programme grant "Digital twins for improved dynamic design", EP/R006768/1 (NG, MDA, SF) The funders had no role in study design, data collection and analysis, decision to publish, or preparation of the manuscript. The authors would like to thank EPSRC, ESRC and MRC for their continued support.

**Competing interests:** The authors have declared that no competing interests exist.

global schools closed [5, 6]. These analyses have highlighted from the beginning that the eventual relaxation of lockdown measures would be problematic [3]. Without a considered cessation of the suppression strategies the risk of a second wave becomes significant, possibly of greater magnitude than the first as the SARS-CoV-2 virus is now endemic in the population [7, 8].

Although much attention was focused on the number of tests being conducted and the effect that testing could have in supressing the disease [9–11]. Not enough attention has been given to the issues of imperfect testing, beyond Matt Hancock, UK Secretary of State for Health and Social Care, stating in a press conference on 2nd April 2020 that "No test is better than a bad test" [12]. In this paper we will explore the validity of this claim.

The failure to detect the virus in infected patients can be a significant problem in high-throughput settings operating under severe pressure, with evidence suggesting that this is indeed the case [13–17]. The public are rapidly becoming aware of the difference between the 'have you got it?' tests for detecting active cases, and the 'have you had it?' tests for the presence of antibodies, which imply some immunity to COVID-19. What may be less obvious is that these different tests need to maximise different test characteristics.

To be useful in ending lockdown measures, active viral tests need to maximise the sensitivity. High sensitivity reduces the chance of missing people who have the virus who may go on to infect others. There is an additional risk that an infected person who has been incorrectly told they do not have the disease, when in fact they do, may behave in a more reckless manner than if their disease status were uncertain.

The second testing approach, seeking to detect the presence of antibodies to identify those who have had the disease would be used in a different strategy. This strategy would involve detecting those who have successfully overcome the virus, and are likely to have some level of immunity (or at least reduced susceptibility to more serious illness if they are infected again), so are relatively safe to relax their personal lockdown measures. This strategy would require a high test specificity, aiming to minimise how often the test tells someone they have had the disease when they haven't [18]. A false positive tells people they have immunity when they don't, which may be worse than if people are uncertain about their viral history.

## Evidence testing is flawed

The successes of South Korea, Singapore, Taiwan and Hong Kong in limiting the impact of the SARS-CoV-2 virus has been attributed to their ability to deploy widespread testing, with digital surveillance, and impose targeted quarantines in some cases [13]. This testing has predominantly been based on the use of reverse transcription polymerase chain reaction (RT-PCR) testing. During the 2009 H1N1 pandemic the rapid development of high sensitivity PCR assay were employed early with some success in that global pandemic [19]. These tests, when well targeted, clearly provide a useful tool for managing and tracking pandemics.

These tests form the basis of much of the research into the incidence, dynamics and comorbidities of SARS-CoV-2, but few, if any, of these studies give consideration to the impact of false test results [20–24]. Increasing reliance on lower-sensitivity tests to address capacity concerns is likely to make available data on confirmed cases more difficult to accurately utilise [19]. It may be the case that false test results contribute to some of the counter-intuitive disease dynamics observed [25].

There is evidence that both active infection [26–30] and antibody [31–33] tests lack perfect sensitivity and specificity even in best-case scenarios. Alternative screening methods such as chest x-rays may be found to have high sensitivity based on biased data [34] or may simply perform poorly even compared to imperfect RT-PCR tests [29]. The Foundation for Innovative

New Diagnostics (FIND) conducted an independent evaluation of five RT-PCR tests which scored highly out of 17 candidate tests on criteria such as regulatory status and availability [35]. Even ideal laboratory conditions can produce a specificity which could be as low as 90%, and the practical specificity is likely to be lower still.

The rapid development and scaling of new diagnostic systems invites error, particularly as labs are converted from other purposes and technicians are placed under pressure, and variation in test collection quality, reagent quality, sample preservation and storage, and sample registration and provenance. Assessing the magnitude of these errors on the performance of tests is challenging in real time. Point-of-care tests are not immune to these errors and are often seen as less accurate than laboratory based tests [36, 37].

## Introduction to test statistics: What makes a 'good' test?

In order to answer this question there are a number of important statistics:

- **Sensitivity** $\sigma$—Out of those who actually have the disease, that fraction that received a positive test result.

- **Specificity** $\tau$—Out of these who did not have the disease, the fraction that received a negative test result.

The statistics that characterise the performance of the test are computed from a confusion matrix (Table 1). We test $n_{infected}$ people who have COVID-19, and $n_{healthy}$ people who do not have COVID-19. In the first group, $a$ people correctly test positive and $c$ falsely test negative. Among healthy people, $b$ will falsely test positive, and $d$ will correctly test negative.

From this confusion matrix the sensitivity is given by (1) and the specificity by (2).

$$\sigma = \frac{a}{n_{infected}} \tag{1}$$

$$\tau = \frac{d}{n_{healthy}}. \tag{2}$$

Sensitivity is the ratio of correct positive tests to the total number of infected people involved in the study characterising the test. The specificity is the ratio of the correct negative tests to the total number of healthy people. Importantly, these statistics depend only on the test itself and do not depend on the population the test is intended to be used upon.

When the test is used for diagnostic purposes, the characteristics of the population being tested become important for interpreting the test results. To interpret the diagnostic value of a positive or negative test result the following statistics must be used:

- **Prevalence** $P$—The proportion of people in the target population that have the disease tested for.

- **Positive Predictive Value** $PPV$—How likely one is to have the disease given a positive test result.

**Table 1. Confusion matrix.**

|  | Infected | Not Infected | Total |
|---|---|---|---|
| Tested Positive | $a$ | $b$ | $a + b$ |
| Tested Negative | $c$ | $d$ | $c + d$ |
| Total | $a + c = n_{infected}$ | $b + d = n_{healthy}$ | $N$ |

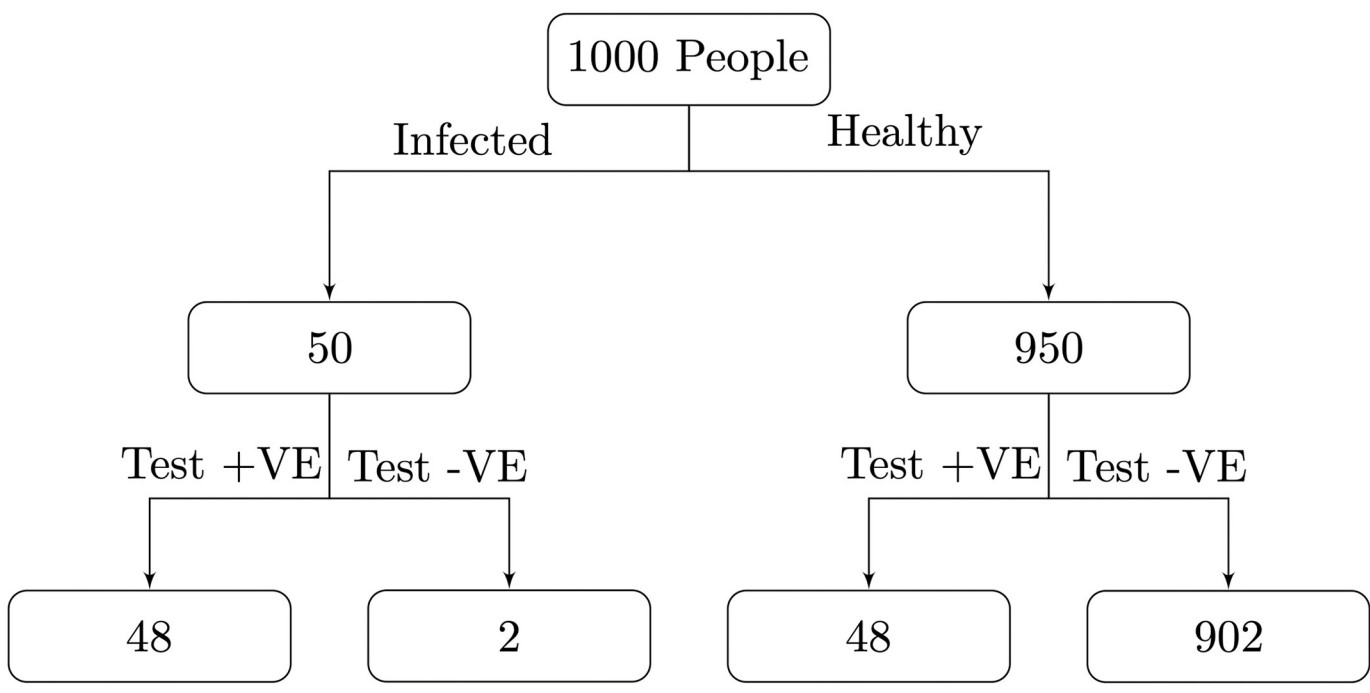

**Fig 1. If the prevalence of a disease amongst those being tested is 0.05 then with $\sigma = \tau = 0.95$ the number of false positives will outnumber the true positives, resulting in $PPV = 0.5$.**

- **Negative Predictive Value** *NPV*—How likely one is to *not* have the disease, given a negative test result.

The *PPV* and *NPV* depend on the prevalence, and hence depend on the population you are focused on. This may an entire nation or region, a sub-population with COVID-19 compatible symptoms, or any other population you may wish to target. The *PPV* and *NPV* can be calculated using Bayes' rule:

$$PPV = \frac{P\sigma}{P\sigma + (1-P)(1-\tau)}, \tag{3}$$

$$NPV = \frac{\tau(1-P)}{\tau(1-P) + (1-\sigma)P}. \tag{4}$$

To illustrate the impact of prevalence on *PPV*, for a test with $\sigma = \tau = 0.95$, if prevalence $P = 0.05$, then the $PPV = 0.5$. Therefore, a positive result only indicates a 50% chance that an individual will have the disease given that they have tested positive, even though the test is highly accurate. Fig 1 shows why, for 1000 test subjects there will be similar numbers of true and false positives even with high sensitivity and specificity of 95%. In contrast, using the same tests on a sample with a higher prevalence $P = 0.5$ we find the $PPV = 0.95$, see Fig 2. Similarly, the *NPV* is lower when the prevalence is higher.

## SIR model with testing

SIR models offer one approach to explore infection dynamics, and the prevalence of a communicable disease. In the generic SIR model, there are *S* people susceptible to the illness, *I* people

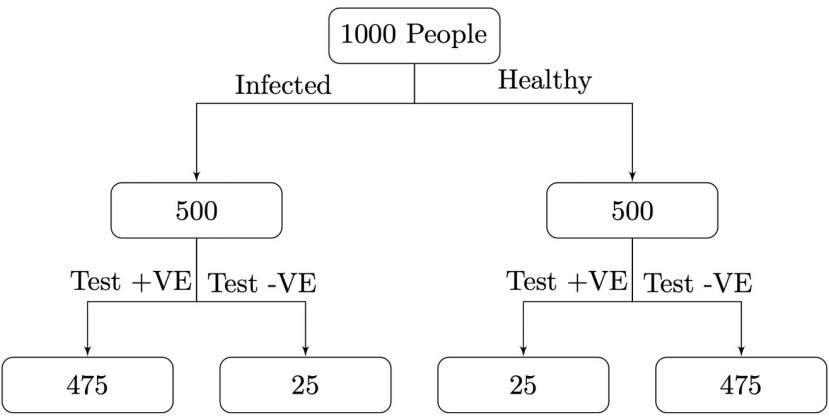

**Fig 2. If the prevalence of a disease amongst those being tested is 0.50 then with $\sigma = \tau = 0.95$ the number of true positives will outnumber the number of false positives, resulting in a high *PPV* of 0.95.**

infected, and *R* people who are recovered with immunity. The infected people are able to infect susceptible people at rate $\beta$ and they recover from the disease at rate $\gamma$ [38], Fig 3 shows how people move between the different states of an SIR model. Once infected persons have recovered from the disease they are unable to become infected again or infect others. This may be because they now have immunity to the disease or because they have unfortunately died.

$$R_0 = \frac{\beta}{\gamma} \tag{5}$$

$$\delta_{S,I} = \beta I S \tag{6a}$$

$$\delta_{I_R} = \gamma I \tag{6b}$$

$$\Delta S = -\delta_{S,I} \tag{6c}$$

$$\Delta I = \delta_{S,I} - \delta_{I,R} \tag{6d}$$

$$\Delta R = \delta_{I,R} \tag{6e}$$

To explore the effect of imperfect testing on the disease dynamics when strategies testing regimes are employed to relax lockdown measures, three new classes were added to the model. The first is a quarantined susceptible state, $Q_S$, the second is a quarantined infected state, $Q_I$, and the third is people who have recovered but are in quarantine, $Q_R$, as shown in Fig 4.

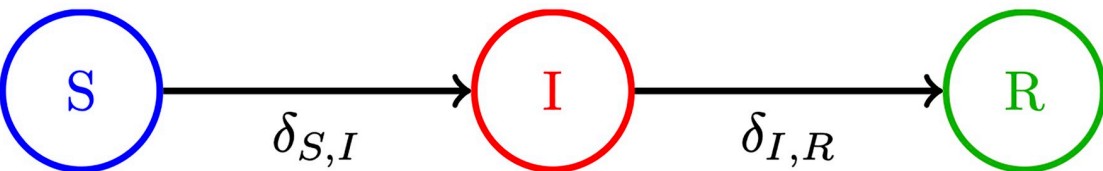

**Fig 3. Diagram for a basic SIR model.** The black arrows show how people move between the different states.

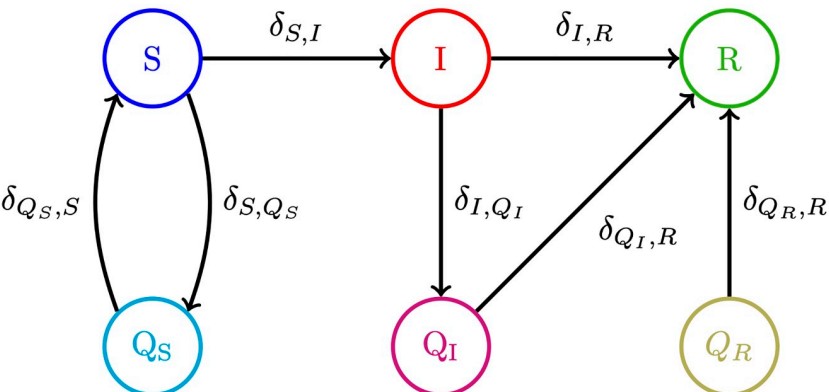

**Fig 4. SIRQ model used to simulated the effect of mass testing to leave quarantine.**

The present model is similar to other SIR models that take into account the effect of quarantining regimes on disease dynamics, such as Lipsitch et al. (2003) [39] or Giordano et al. (2020) [23]. Lipsitch et al. implement quarantine in their model but do not incorporate the effects on the dynamics from imperfect testing, nor do they consider how the quality and scale of an available test affect the spread of a disease. Diagnostic uncertainty plays no part in the model they present. Likewise, Giordano et al reduce population based diagnostic strategies to two parameters which confound test capacity, test targeting, and diagnostic uncertainty. Again, they do not investigate the role that diagnostic uncertainty plays in the spread of a disease. The intent of this model is not to create a more sophisticated SIR model, but to investigate how diagnostic uncertainty affects the dynamics of an epidemic.

The model evaluates each day's population-level state transitions. There are two possible tests that can be performed:

- An active virus infection test that is able to determine whether or not someone is currently infectious. This test is performed on some proportion of the un-quarantined population ($S + I + R$). It has a sensitivity of $\sigma_A$ and a specificity of $\tau_A$.

- An antibody test that determines whether or not someone has had the infection in the past. This is used on the fraction of the population that is currently in quarantine but not infected ($Q_S + Q_R$) to test whether they have had the disease or not. This test has a sensitivity of $\sigma_B$ and a specificity of $\tau_B$.

Each test is defined by a number of parameters. Testing each day is limited by the test capacity $C$, the maximum number of tests that can be performed each day. Each day a population $N$ will be submitted for testing. The targeting capability of the test, $T$ indicates the probability that an individual submitted for testing is positive, this is effectively the PPV of the initial screening effort. This results in a number of individuals $M$ being considered for screening who are negative, of which $K$ will be tested. Targeting must be imperfect, as if it were perfect there would be no need for testing. Unless otherwise stated, scenarios consider a default targeting of $T = 0.8$, representing an extremely effective screening capability that is nonetheless imperfect.

If daily testing targets are a goal regardless of the prevalence of the illness, $T$ can be overruled to ensure $N \approx C$ for example. This condition is referred to as Strict Capacity and is denoted with boolean parameter $G$, defaulting to true for all scenarios. Tests can also be conducted periodically by changing the test interval parameter $D$. These default to 1, i.e. daily testing.

Each test has unique parameters, so for example test A (active virus infection test) has a targeting parameter $T_A$ whilst test B (antibody test) has $T_B$. The parameters $\sigma$, $\tau$, $T$, $C$, $G$ and $D$ define a test.

A person in any category who tests positive in an active virus test transitions into the corresponding quarantine state, where they are unable to infect anyone else. A person, in $Q_S$ or $Q_R$, who tests positive in an antibody test transitions to $S$ and $R$ respectively. Any person within $I$ or $Q_I$ who recovers transitions to $R$, on the assumption that the end of the illness is clear and they will know when they have recovered.

For this parameterisation the impact of being in the susceptible quarantined state, $Q_S$, makes an individual insusceptible to being infected. Similarly, being in the infected quarantined state, $Q_I$, individuals are unable to infect anyone else. In practicality there is always leaking, no quarantine is entirely effective, but for the sake of exploring the impact of testing uncertainty these effects are neglected from the model. Other situations may require including this effect.

The SIR model used in this paper uses discrete-time binomial sampling for calculating movements of individuals between states. For a defined testing strategy these rates are defined as follows:

$$M_A = \min\left(S, C_A - I, \max\left(0, \frac{I}{T_A} - I, C_A - I\right)\right) \tag{7a}$$

$$N_A = \min(C_A, M_A + I) \tag{7b}$$

$$K_A = \mathrm{H}(M_A, I, N_A) \tag{7c}$$

$$\delta_{S,Q_S} = \mathrm{Bin}(K_A, 1 - \tau_A) \tag{7d}$$

$$\delta_{I,Q_I} = \mathrm{Bin}(N_A - K_A, \sigma_A) \tag{7e}$$

$$\delta_{S,I} = \min\left(S - \delta_{S,Q_S}, \mathrm{Bin}\left(I, \frac{\beta(S - \delta_{S,Q_S})}{S + I + R - \delta_{I,Q_I} - \delta_{S,Q_S}}\right)\right) \tag{7f}$$

$$\delta_{I,R} = \mathrm{Bin}(I - \delta_{I,R}, \gamma) \tag{7g}$$

$$M_B = \min\left(Q_S, C_B - Q_R, \max\left(0, \frac{Q_R}{T_B} - Q_R, C_B - Q_R\right)\right) \tag{7h}$$

$$N_B = \min(C_B, M_B + Q_R) \tag{7i}$$

$$K_B = \mathrm{H}(M_A, I, N_A) \tag{7j}$$

$$\delta_{Q_S,S} = \mathrm{Bin}(K_B, 1 - \tau_B) \tag{7k}$$

$$\delta_{Q_I,R} = \mathrm{Bin}(Q_R, \gamma) \tag{7l}$$

$$\delta_{Q_R,R} = \text{Bin}(N_B - K_B, \sigma_B) \tag{7m}$$

$$\Delta S = \delta_{Q_S,S} - \delta_{S,Q_S} - \delta_{S,I} \tag{7n}$$

$$\Delta I = \delta_{S,I} - \delta_{I,Q_I} - \delta_{I,R} \tag{7o}$$

$$\Delta R = \delta_{I,R} + \delta_{Q_I,R} + \delta_{Q_R,R} \tag{7p}$$

$$\Delta Q_S = \delta_{S,Q_S} - \delta_{Q_S,S} \tag{7q}$$

$$\Delta Q_I = \delta_{I,Q_I} - \delta_{Q_I,R} \tag{7r}$$

$$\Delta Q_R = -\delta_{Q_R,R} \tag{7s}$$

In Eq 7, Bin($n$, $p$) refers to a binomial distribution with count $n$ and rate $p$, H($n$, $k$, $m$) refers to a hypergeometric distribution with populations $n$ and $k$ and a sample size $m$.

The model must be initialised with a defined population split between the six states. At each time step $t$, the model calculates the number of persons moving between each state in the order defined above. The use of binomial and hypergeometric sampling was prompted by a desire to incorporate aleatory uncertainty in each movement. The current approach does not account for epistemic uncertainty, fixing the model parameters $\sigma$, $\tau$, $C$, $T$ and $D$. A discrete time model was selected to allow for comparisons against available published data detailing recorded cases and recoveries on a day-by-day basis.

If the tests were almost perfect, then we can imagine how the epidemic would die out very quickly by either widespread infection or antibody testing with a coherent management strategy. A positive test on the former and the person is removed from the population, and positive test on the latter and the person, unlikely to contract the disease again, can join the population.

More interesting are the effects of incorrect test results on the disease dynamics. If someone falsely tests positive in the antibody test, they enter the susceptible state. Similarly, if an infected person receives a false negative for the disease they remain active in the infected state and hence can continue the disease propagation and infect further people.

## What part will testing play in relaxing lockdown measures?

In order to explore the possible impact of testing strategies on the relaxation of lockdown measures several scenarios have been analysed. These scenarios are illustrative of the type of impact, and the likely efficacy of a range of different testing configurations.

- **Immediate end to lockdown scenario**: This baseline scenario is characterised by a sudden relaxation of lockdown measures.

- **Immunity passports scenario**: A policy that has been discussed in the media [40–42]. Analogous to the International Certificate of Vaccination and Prophylaxis, antibody based testing would be used to identify those who have some level of natural immunity.

- **Incremental relaxation scenario**: A phased relaxation of lockdown is the most likely policy that will be employed. To understand the implications of such an approach this scenario has explored the effect of testing capacity and test performance on the possible disease dynamics

under this type of policy. Under the model parameterisation this analysis has applied an incremental transition rate from the $Q_S$ state to the $S$ state, and $Q_R$ to $R$.

Whilst the authors are sensitive to the sociological and ethical concerns of any of these approaches, the analysis presented is purely on the question of efficacy.

For the purpose of the analysis we have selected a population similar in size to the United Kingdom, $6.7 \times 10^7$ people, $\beta$ and $\gamma$ were set to 0.32 and 0.1 respectively, this was ensure that $R_0$ value of the model was broadly in line with other models [43, 44].

### Immediate end to lockdown scenario

Under the baseline scenario, characterised by the sudden and complete cessation of lockdown measures, we explored the impact of infection testing. Under this formulation the initial conditions of the model in this scenario is that the all of the population in $Q_S$ transition to $S$ in the first iteration. The impact of infection testing under this scenario was analysed in Fig 5 using the parameters shown in Table 2.

These scenarios consider the impact of attempts to control the disease through increased testing capacity and a more sensitive test. A test capacity range between $1 \times 10^5$ and $2 \times 10^5$ was considered as representative of the capabilities of a country such as the UK. To illustrate the sensitivity of the model to testing scenarios an evaluation was conducted with a range of infection test sensitivities, from 50% (i.e of no diagnostic value) to 98%. The specificity of these tests has a negligible impact on the disease dynamics in these scenarios. A false positive would mean people are unnecessarily removed from the susceptible population, but the benefit of a reduction in susceptible population is negligibly small.

As would be expected the model indicates a second wave is an inevitability and as many as 20 million people could become infected within 30 days. A high-sensitivity test has little impact

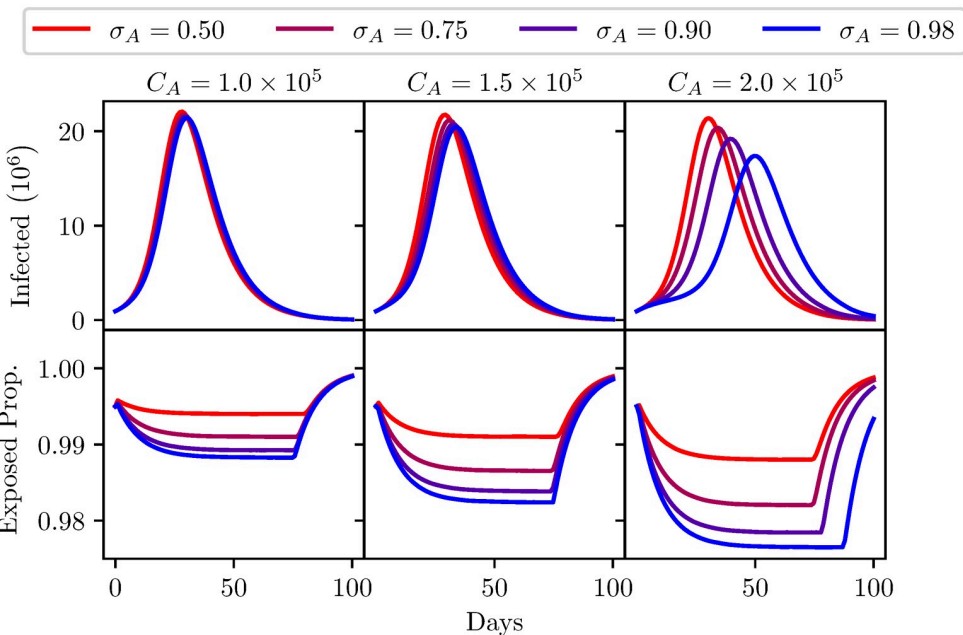

**Fig 5. A comparison of different infection test sensitivities $\sigma_A$ shown from red to blue.** Three different infection test capacities are considered. Left: test capacity = $1 \times 10^5$. Centre: test capacity = $1.5 \times 10^5$. Right: test capacity = $2 \times 10^5$. Top: The number of infected individuals ($I + Q_I$ population) over 100 days. Bottom: The proportion of the population that has been released from quarantine ($S + I + R$ population) over 100 days. Model parameters are shown in Table 2.

**Table 2. Fixed parameters used for Fig 5 analysis.** Antibody tests were disabled for this analysis.

| Model Parameters | | | | | | |
|---|---|---|---|---|---|---|
| $\sigma_A$ | $\tau_A$ | $T_A$ | $C_A$ | $G_A$ | $\beta$ | $\gamma$ |
| - | 0.9 | 0.8 | - | True | 0.32 | 0.1 |
| Initial Population split | | | | | | |
| Population | $S$ | $I$ | $R$ | $Q_S$ | $Q_I$ | $Q_R$ |
| $6.7 \times 10^7$ | 0.984 | 0.01 | 0.001 | 0 | 0.004 | 0.001 |

beyond quarantining a slightly higher percentage of the population if capacities are low. At higher capacities this patterns remains, though peak infection counts are marginally reduced. Overall it is clear that reliance on infection testing, even with a highly sensitive test and high capacities, is not enough to prevent widespread infection.

## Immunity passports scenario

The immunity passport is an idiom describing an approach to the relaxation of lockdown measures that focuses heavily on antibody testing. Wide-scale screening for antibodies in the general population promises significant scientific value, and targeted antibody testing is likely to have value for reducing risks to NHS and care-sector staff, and other key workers who will need to have close contact with COVID-19 sufferers. The authors appreciate these other motivations for the development and roll-out of accurate antibody tests. This analysis however focuses on the appropriateness of this approach to relaxing lockdown measures by mass testing the general population. Antibody testing has been described as a 'game-changer' [45]. Some commentators believe this could have a significant impact on the relaxation of lockdown measures [41], but others note that there are severe ethical, logistical and medical concerns which need to be resolved before antibody testing could support a strategy such as this [46].

Much of the discussion around antibody testing in the media has focused on the performance and number of these tests. The efficacy of this strategy however is far more dependent on the prevalence of antibodies (seroprevalence) in the general population. Without wide-scale antibody screening it is impossible to know the seroprevalence in the general population, so there is scientific value in such an endeavour. However, the seroprevalence is the dominant factor to determine how efficacious antibody screening would be for relaxing lockdown measures.

Presumably, only people who test positive for antibodies would be allowed to leave quarantine. The more people in the population with antibodies, the more people will get a true positive, so more people would be correctly allowed to leave quarantine (under the paradigms of an immunity passport).

The danger of such an approach are false positives. We demonstrate the impact of people reentering the susceptible population who have no immunity. We assume their propensity to contract the infection is the same as those without the false sense of security a positive test may engender. On an individual basis, and even at the population level, behavioural differences between those with false security from a positive antibody test, versus those who are uncertain about their viral history could be significant. The model parametrisation here does not include this additional confounding effect.

To simulate the seroprevalence in the general population the model is preconditioned with different proportions of the population in the $Q_S$ and $Q_R$ states. This is analogous to the proportion of people that are currently in quarantine who have either had the virus and developed some immunity, and the proportion of the population who have not contracted the virus and

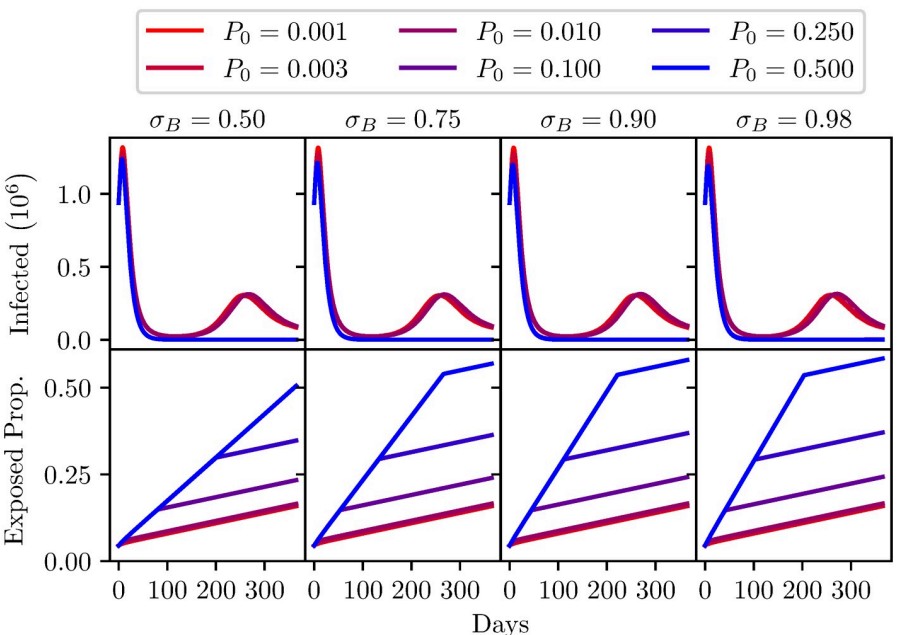

**Fig 6. A comparison of different antibody test sensitivities $\sigma_B$, with varying levels of seroprevalence ($P_0$).** Top: The number of infected individuals ($I + Q_I$ population) over one year. Bottom: The proportion of the population that has been released from quarantine ($S + I + R$ population) over one year. Model parameters are shown in Table 3.

have no immunity. Of course the individuals in these groups do not really know their viral history, and hence would not know which state they begin in. The model evaluations explore a range of sensitivity and specificities for the antibody testing. These sensitivity and specificities, along with the capacity for testing, govern the transition of individuals from $Q_R$ to $R$ (true positive tests), and from $Q_S$ to $S$ (false positive tests).

Figs 6 and 7 show the results of the model evaluations, the parameters for these runs are shown in Tables 3 and 4. The top row of each figure corresponds to the number of infections in time, the bottom row of each figure is the proportion of the population that are released from quarantine and hence are now in the working population. Maximising this rate of reentry into the population is of course desirable, and it is widely appreciated that some increase in the numbers of infections is unavoidable. The desirable threshold in the trade-off between societal activity and number of infections is open to debate.

Each of the plots in Figs 6 and 7 show the effect of different seroprevalence in the population. To be clear, this is the proportion of the population that has contracted the virus and recovered but are in quarantine. The analysis has explored a range of seroprevalence from 0.1% to 50%. Fig 6 explores the impact of a variation in sensitivity, from a test with 50% sensitivity to tests with a high sensitivity of 98%.

It can be seen, considering the top row of Fig 6, that the sensitivity of the test has no discernible impact on the number of infections. The seroprevalence entirely dominates. This is possibly counter intuitive, but as was discussed above, even a highly accurate test produces a very large number of false positives when seroprevalence is low. In this case that would mean a large number of people are allowed to re-enter the population, placing them at risk, with a false sense of security that they have immunity.

The bottom row of Fig 6 shows the proportion of the entire population leaving quarantine over a year of employing this policy. At low seroprevalence there is no benefit to better

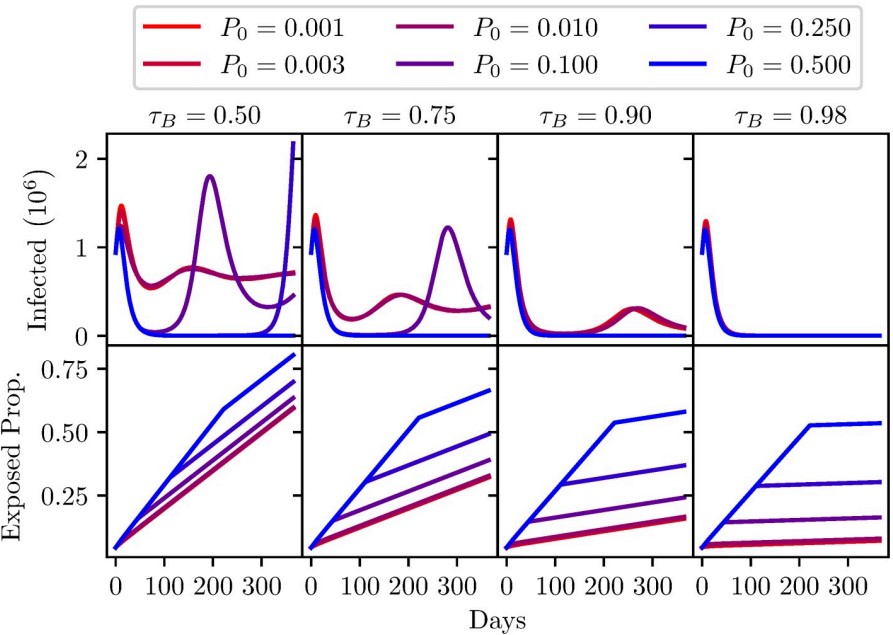

**Fig 7. A comparison of different antibody test specificities $\tau_B$ shown from left to right, with varying levels of seroprevalence ($P_0$) shown from red to blue.** Top: The number of infected individuals ($I + Q_I$ population) over one year. Bottom: The proportion of the population that has been released from quarantine ($S + I + R$ population) over one year. Model parameters are shown in Table 4.

performing tests. This again may seem obscure to many readers. If you consider the highest seroprevalence simulation, where 50% of the population have immunity, higher sensitivity tests are of course effective at identifying those who are immune, and gets them back into the community much faster.

A more concerning story can be seen when considering the graphs in Fig 7. Now we consider a range of antibody test specificities. Going from 50% to 98%. Low specificities ($\tau < 0.9$)

**Table 3. Fixed parameters used for Fig 6 analysis.** Infection tests were disabled for this analysis.

| Model Parameters | | | | | | |
|---|---|---|---|---|---|---|
| $\sigma_B$ | $\tau_B$ | $T_B$ | $C_B$ | $G_B$ | $\beta$ | $\gamma$ |
| - | 0.9 | 0.8 | $2 \times 10^5$ | True | 0.32 | 0.1 |
| Initial Population split | | | | | | |
| Population | $S$ | $I$ | $R$ | $Q_S$ | $Q_I$ | $Q_R$ |
| $6.7 \times 10^7$ | 0.035 | 0.01 | 0.001 | $0.95(1 - P_0)$ | 0.004 | $0.95 P_0$ |

**Table 4. Fixed parameters used for Fig 7 analysis.** Infection tests were disabled for this analysis.

| Model Parameters | | | | | | |
|---|---|---|---|---|---|---|
| $\sigma_B$ | $\tau_B$ | $T_B$ | $C_B$ | $G_B$ | $\beta$ | $\gamma$ |
| 0.9 | - | 0.8 | $2 \times 10^5$ | True | 0.32 | 0.1 |
| Initial Population split | | | | | | |
| Population | $S$ | $I$ | $R$ | $Q_S$ | $Q_I$ | $Q_R$ |
| $6.7 \times 10^7$ | 0.035 | 0.01 | 0.001 | $0.95(1 - P_0)$ | 0.004 | $0.95 P_0$ |

lead to extreme second peaks, and could possibly lead to more. This is due to the progressive release of false-positives from the quarantined population, which eventually swells the susceptible population to a size where the infection count can resume exponential growth. High specificities avoid this at the cost of a prolonged lockdown, which is naturally limited by the lower false-positive rate. Clearly some means of release beyond immunity passports would be required to avoid this scenario. Notably, a reasonably specific test ($\tau_B = 0.9$) is capable of restraining a second peak to reasonably low levels regardless of seroprevalence. This may allow for other means of reducing lockdown measures, though with very low seroprevalence this could still be a potentially risky strategy. The dangers of neglecting uncertainties in medical diagnostic testing are pertinent to this decision [47].

### Incremental relaxation scenario

Considering the above, some form of incremental relaxation of lockdown seems appropriate. This could take many forms, it could be an incremental restoration of certain activities such as school openings, permission for the reopening of some businesses, the relaxation of stay-at-home messaging, etc. Under the parameterisation chosen for this analysis the model is not sensitive to any particular policy change. We consider a variety of rates of phased relaxations to quarantine. To model these rates we consider a weekly incremental transition rate from $Q_S$ to $S$, and $Q_R$ to $R$. In Fig 8, three weekly transition rates have been applied: 1%, 5% and 10% of the quarantined population. Whilst in practice the rate is unlikely to be uniform as decision makers would have the ability to update their timetable as the impact of relaxations becomes apparent, it is useful to illustrate the interaction of testing capacity and release rate.

The model simulates these rates of transition for a year, with a sensitivity and specificity of 90% for active virus tests. The specifics of all the runs are detailed in Table 5. Fig 8 shows five analyses, with increasing capacity for the active virus tests. In each, the 3 incremental transition rates are applied with a range of targeting capabilities. The value of 0.8 used previously represents an unrealistically extreme case of effective targeting. The *PPV*, as discussed above, has a greater dependence on the prevalence (at lower values) in the tested population than it does on the sensitivity of the tests, the same is true of the specificity and the *NPV*.

It is important to notice that higher test capacities cause a higher peak of infections for higher release rates. This has a counterintuitive explanation. When there is the sharpest rise in the susceptible population (i.e., high rate of transition), the virus rapidly infects a large number of people. When these people recover after around two weeks they become immune and thus cannot continue the spread of the virus. However, when the infection testing is conducted with a higher capacity up to 150,000 units per day, these tests transition some active viral carriers into quarantine, so the peak is slightly delayed providing more opportunity for those released from quarantine later to be infected, leading to higher peak infections. This continues until the model reaches effective herd immunity after which the number of infected in the population decays very quickly. Having higher testing capacities delays but actually has the potential to worsen the peak number of infections.

At 10% release rate, up to a capacity of testing of 150,000 these outcomes are insensitive to the prevalence of the disease in the tested population. This analysis indicates that the relatively fast cessation of lockdown measures and stay-home advice would lead to a large resurgence of the virus. Testing capacity of the magnitude stated as the goal of the UK government would not be sufficient to flatten the curve in this scenario.

The 1% release rate scenario indicates that a slow release by itself is sufficient to lower peak infections, but potentially extends the duration of elevated infections. The first graph of the top row in Fig 8 shows that the slow release rate causes a plateau at a significantly lower

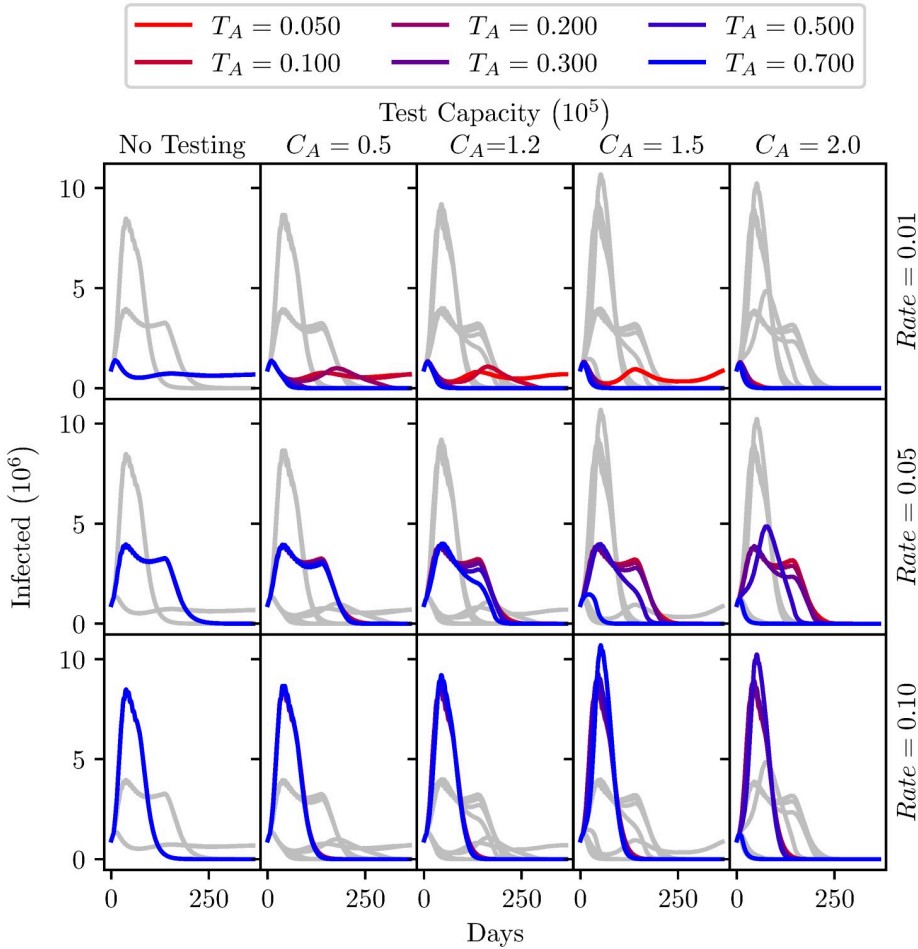

**Fig 8. Total active infections each day over the year after relaxing lock-down, under different testing intensities (columns) and various epidemiologic conditions.** The per-day testing capacity is varied across the five columns of graphs. Rate, the percentage of the initial quarantined population being released each week is varied among rows. The prevalence of infections in the tested population is varied among different colours. To facilitate comparison within each column of graphs, the gray curves show the results observed for other Rates and Prevalences with the same testing intensity. Model parameters are shown in Table 5.

number of infections compared to the other release rates. Poorly targeted tests at capacities less than 100,000 show similar consistent levels of infections. However, with a targeted test having a prevalence of 30% or more, the 1% release rate indicates that even with 50,000 tests per day continuous suppression of the infection may be possible.

**Table 5. Fixed parameters used for Fig 8 analysis.**

| Model Parameters | | | | | | | |
|---|---|---|---|---|---|---|---|
| $\sigma_A$ | $\tau_A$ | $T_A$ | $C_A$ | $G_A$ | $D_A$ | $\beta$ | $\gamma$ |
| 0.9 | 0.9 | - | - | True | 1 | 0.32 | 0.1 |
| $\sigma_B$ | $\tau_B$ | $T_B$ | $C_B$ | $G_B$ | $D_B$ | | |
| 1 | 0 | 0 | $Rate \times Population$ | True | 7 | | |
| Initial Population split | | | | | | | |
| Population | $S$ | $I$ | $R$ | $Q_S$ | $Q_I$ | $Q_R$ | |
| $6.7 \times 10^7$ | 0.034 | 0.01 | 0.001 | 0.95 | 0.004 | 0.001 | |

At the rate of 5% of the population in lock-down released incrementally each week the infection peak is suppressed compared to the 10% rate. The number of infections would remain around this level for a significantly longer period of time, up to 6 months. There is negligible impact of testing below a capacity of 100,000 tests. However, with a test capacity of 150,000 tests the duration of the elevated levels of infections could be reduced if the test is extremely well targeted ($T_A$ = 0.7), reducing the length of necessary wide-scale lockdown. If this level of targeting is not achieved, increasing capacity may again increase peak infections, so care must be taken to ensure a highly targeted testing strategy.

## Conclusions

This analysis does support the assertion that a bad test is potentially worse than no tests, but a good test is only effective in a carefully designed strategy. More is not necessarily better and over estimation of the test accuracy could be extremely detrimental.

This analysis is not a prediction; the numbers used in this analysis are estimates and the SIRQ model used is unlikely to be detailed enough to inform policy decisions. As such, the authors are not drawing firm conclusions about the absolute necessary capacity of tests. Nor do they wish to make specific statements about the necessary sensitivity or specificity of tests or the recommended rate of release from quarantine. The authors do, however, propose some conclusions that would broadly apply when testing and quarantining regimes are used to suppress epidemics, and therefore believe they should be considered by policy makers when designing strategies to tackle COVID-19.

- Diagnostic uncertainty can have a large effect on the dynamics of an epidemic. And, sensitivity, specificity, and the capacity for testing alone are not sufficient to design effective testing procedures. Policy makers need to be aware of the accuracy of the tests, the prevelence of the disease at increased granularity and the characteristics of the target population, when deciding on testing strategies.

- Caution should be exercised in the use of antibody testing. Assuming that the prevalence of antibodies is low, it is unlikely antibody testing at any scale will support the end of lockdown measures. And, un-targeted antibody screening at the population level could cause more harm than good.

- Antibody testing, with a high specificity may be useful on an individual basis, it has scientific value, and could reduce risk for key workers. But any belief that these tests would be useful to relax lockdown measures for the majority of the population is misguided.

- The incremental relaxation to lockdown measures, with all else equal, would significantly dampen the increase in peak infections, by 1 order of magnitude with a faster relaxation, and 2 orders of magnitude with a slower relaxation.

- As the prevalence of the disease is suppressed in different regions, it may be the case that small spikes in cases could be the result of false positives. This problem is potentially exacerbated by increased testing in localities in response to small increases in positive tests. Policy decisions that depend on small changes in the number of positive tests may, therefore, be flawed.

- For infection screening to be used to relax quarantine measures the capacity needs to be sufficiently large but also well targeted to be effective. For example this could be achieved through effective contact tracing. Untargeted mass screening at any capacity would be ineffectual and may prolong the necessary implementation of lockdown measures.

Epidemiological models used for policy making in real time will need to take into account the impact of diagnostic uncertainty of testing, as well as the dynamical behaviour and sensitivity analyses of modelled parameters in an appropriately complex model that may need to include quarantining, contact tracing and other surveillance strategies, test availability and targeting, and multiple subpopulations of susceptible, infected and recovered categories.

## Author Contributions

**Conceptualization:** Nicholas Gray, Dominic Calleja, Alexander Wimbush, Enrique Miralles-Dolz, Ander Gray, Marco De Angelis, Elfriede Derrer-Merk, Bright Uchenna Oparaji, Vladimir Stepanov, Louis Clearkin, Scott Ferson.

**Formal analysis:** Dominic Calleja, Alexander Wimbush.

**Investigation:** Nicholas Gray.

**Methodology:** Nicholas Gray, Dominic Calleja, Alexander Wimbush, Enrique Miralles-Dolz, Ander Gray, Marco De Angelis.

**Supervision:** Nicholas Gray.

**Writing – original draft:** Nicholas Gray, Dominic Calleja, Alexander Wimbush, Enrique Miralles-Dolz, Ander Gray, Marco De Angelis, Elfriede Derrer-Merk, Bright Uchenna Oparaji, Vladimir Stepanov, Louis Clearkin, Scott Ferson.

**Writing – review & editing:** Nicholas Gray, Dominic Calleja, Alexander Wimbush, Enrique Miralles-Dolz, Ander Gray, Marco De Angelis, Elfriede Derrer-Merk, Bright Uchenna Oparaji, Vladimir Stepanov, Louis Clearkin, Scott Ferson.

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
