## [Decision Letter · Decision Letter 0]

7 Jul 2020

PONE-D-20-13370

"No test is better than a bad test'': Impact of diagnostic uncertainty in mass testing on the spread of COVID-19

PLOS ONE

Dear Dr. Gray,

Thank you for submitting your manuscript to PLOS ONE. After careful consideration, we feel that it has merit but does not fully meet PLOS ONE’s publication criteria as it currently stands. Therefore, we invite you to submit a revised version of the manuscript that addresses the points raised during the review process.

The manuscript is timely, and the reviewers agreed with the overall conclusions of the manuscript. However, they raised a few key concerns that should be addressed in a revised manuscript.

We look forward to receiving your revised manuscript.

Kind regards,

Jishnu Das, Ph.D.

Academic Editor

PLOS ONE

Journal Requirements:

2. Please provide  the full details of your models in  your Methods section, and not as Supplementary files; and ensure that all parameters have been described in sufficient detail to meet our reproducibility criteria.

3. Thank you for stating the following in your Competing Interests section: 'NO'

a. Please complete your Competing Interests statement to state any Competing Interests. If you have no competing interests, please state "The authors have declared that no competing interests exist.", as detailed online in our guide for authors at http://journals.plos.org/plosone/s/submit-now

5. Please ensure that you refer to Figure 3 in your text as, if accepted, production will need this reference to link the reader to the figure.

Reviewers' comments:

Reviewer's Responses to Questions

**Comments to the Author**

1. Is the manuscript technically sound, and do the data support the conclusions?

Reviewer #1: Partly

Reviewer #2: Yes

2. Has the statistical analysis been performed appropriately and rigorously? 

Reviewer #1: N/A

Reviewer #2: N/A

3. Have the authors made all data underlying the findings in their manuscript fully available?

Reviewer #1: Yes

Reviewer #2: Yes

4. Is the manuscript presented in an intelligible fashion and written in standard English?

Reviewer #1: Yes

Reviewer #2: Yes

5. Review Comments to the Author

Reviewer #1: In this manuscript, the authors set out to explore the important issue of the coupled impact of diagnostic uncertainty, limited testing capacity and various quarantine relaxation strategies on the overall spread of COVID19. Definitions of standard diagnostic quality parameters (e.g. sensitivity, specificity, PPV, NPV) are first reproduced in detail. This is followed by the presentation of a modified simple SIR model including quarantine states. This is then used to model three quarantine relaxation scenarios, using assumed numbers and parameters. Notably, no comparison with real world data is presented. A final conclusion is eventually drawn in favor of slow release of quarantine and targeted use of imperfect tests.

Overall, this manuscript does present some interesting and potentially useful analysis and conclusions. However a significant body of COVID19 epidemiological modeling literature is now available which also considers some of the same questions. This work can thus be significantly improved by placing it better in the context of the existing literature and clearly highlighting the unique aspects of its analysis and conclusions. Some specific comments and suggestions for improvement are below.

Major comments:

1. The title of the article seems to make a provocative claim that could be interpreted as – performing no diagnostic testing at all would be more beneficial than using imperfect diagnostic testing. Given that this is not what is proven by the analysis shown in the rest of the manuscript, this reviewer would strongly recommend modifying the title to drop this rather misleading claim.

2. This article would benefit significantly from placing it in the context of existing COVID19 epidemiological modeling literature. Much more detailed models (Lipsitch et al – DOI: 10.1126/science.1086616; Giordano et al – DOI: 10.1038/s41591-020-0883-7 etc) along with comparisons to real world data are available now. How and why is the arguably more simplistic model presented here still valuable needs to be clearly justified.

3. A critical aspect of almost all current quarantine strategies is isolation not only of those who test positive for the virus but tracing and isolation of their close contacts as well. The effect of this is neglected here. Can the authors comment on how this might affect their conclusions?

4. A large amount of real world data is now available about COVID19. Can the authors test their analysis and conclusions using any of these data sets? The robustness of their conclusions can be significantly improved if even a partial comparison is presented.

5. In lines 280-281, a 10-fold lower viral testing capacity (10,000/day) is used compared to that (100,000/day) used earlier in the article without providing a justification. As of now, it seems the 100,000 number has in fact been surpassed in the UK. Does this change conclusions?

6. An assumed UK context is implicitly used at a number of places in the article. For clarity, these should be explicitly stated – including all numbers of population or parameters used that depend on this context.

Minor comments/suggestions:

1. The amount of detail in which textbook definitions of diagnostic quality parameters is reproduced here – while potentially useful to the lay reader – is not necessary. Citations to relevant texts or other sources can suffice.

2. Similarly, Figure 1 and 2 – to the extent that they are needed at all – would benefit from being converted to a plot showing variation of PPV with prevalence instead.

2. In general, in this reviewer’s opinion, the authors here adopt a more journalistic or colloquial tone in their writing than is usual in scientific literature. A significant fraction of the citations are from popular media as well. All of this only ends up distracting the reader from the scientific content. This is avoidable and can be easily rectified.

Reviewer #2: The developed modified SIR model presents a simple yet powerful model of the dynamics of susceptible, infected and recovered proportions of the population in quarantine or active. Three useful scenarios were tested, and several intuitive or interesting dynamical behaviors were reported. Overall, with some improvements, this manuscript can become more accessible and impactful:

- The availability of the model implementation codes (which platform/language?) would improve the impact by enabling the testing of new strategies for the relaxation of current social distancing measures outside the 3 tested scenarios by the readers. Additionally, use of tables to fully report the parameters that are kept constant in each scenario/plot are required for reproducibility of the results.

- While the temporal plots and the choice of parameters were selected wisely to emphasize the key points of the paper and interesting dynamics, the complex interactions of the key variables of the model call for more rigorous analysis to fully capture the nonlinear dynamics. As hinted in the top left two panels of Figure 6, nonlinear dynamics are observed such as oscillatory and dampening dynamics. These call for additional rigorous analysis such as sensitivity analysis to key parameters of the system in each scenario, or parameter sweeps, phase portraits, or depicting the phenotypic spaces (e.g. key dynamical behaviors in the two-dimensional space of p and tau_B).

- Minor edits/typo:

o Typo in text line 128, PPV=0.95, 0.8 is not correct. (as also noted in Figure caption).

o Although expected from the definition of Bayes formula, I think it would be beneficial to emphasize from the beginning that prevalence means different things in the viral and antibody tests.

6. PLOS authors have the option to publish the peer review history of their article (what does this mean?). If published, this will include your full peer review and any attached files.

Reviewer #1: No

Reviewer #2: **Yes: **Sepideh Dolatshahi

---

## [Author Response · Author response to Decision Letter 0]

28 Aug 2020

We would like to thank you, and the reviewers, for considering our manuscript and suggesting revisions among their helpful comments.

All additional journal requirements have been met.

Reviewer #1 - Comment 1. The title of the article seems to make a provocative claim that could be interpreted as – performing no diagnostic testing at all would be more beneficial than using imperfect diagnostic testing. Given that this is not what is proven by the analysis shown in the rest of the manuscript, this reviewer would strongly recommend modifying the title to drop this rather misleading claim.

Reviewer 1 (R1) takes issue with the title of the manuscript, arguing it is provocative and that the claim “no test is better than a bad test” isn’t borne out in the conclusions of the paper. The title of this manuscript is a direct quote from Matt Hancock, UK Secretary of State for Health and Social Care, and the manuscript itself is intended to investigate this statement. That this has been interpreted as a statement by the authors is a mistake on our part, and we have amended the title and the manuscript to make it more clear that this is a contested statement that we are intending to investigate. Broadly, we do suggest that bad tests are potentially counterproductive if they are used to justify the removal of measures that prevent the spread of a disease, and the analysis bears this out. We do not suggest that no testing at all is the ideal approach, and have made this clear in the revision.

Reviewer #1 - Comment 2. This article would benefit significantly from placing it in the context of existing COVID19 epidemiological modeling literature. Much more detailed models (Lipsitch et al – DOI: 10.1126/science.1086616; Giordano et al – DOI: 10.1038/s41591-020-0883-7 etc) along with comparisons to real world data are available now. How and why is the arguably more simplistic model presented here still valuable needs to be clearly justified.

We believe R1 has not fully understood the research questions we are trying to explore and in this confusion asks for further justification of the simple model we present against more detailed SIR models that also include the dynamics of diagnosis and quarantine strategies. R1 suggests two papers, Lipsitch et al. (2003) and Giordano et al. (2020), as having better models than the one that we employ, however neither of these models would be able to answer the question we are trying to answer.

Lipsitch et al. implement quarantine in their model but do not incorporate the effects on the dynamics from imperfect testing, nor do they consider how the quality and scale of an available test affect the spread of a disease. Diagnostic uncertainty plays no part in the model they present. Likewise, Giordano et al. reduce diagnosis to two parameters, ε and θ, which confound test capacity, test targeting, and diagnostic uncertainty. Again, they do not investigate the role that diagnostic uncertainty plays in the spread of a disease. The analysis presented in our manuscript could be considered an in-depth look into these specific parameters using a simpler model than the SIDARTHE model used by Giordano et al. The intent of our paper is not to create a more sophisticated SIR model, but to investigate how diagnostic uncertainty affects the dynamics of an epidemic.

The value of our simple model is in demonstrating the impact of a higher-quality test (i.e. one with greater sensitivity and specificity) with better targeting (prevalence in the latter case). If such an approach could be incorporated into a more sophisticated model such as SIDARTHE this could certainly prove more informative, but the paper’s contribution is the demonstration that diagnostic uncertainty can have significant effect on epidemics. We have added text that more clearly spells out the contributions and its value within the manuscript.

Reviewer #1 - Comment 3. A critical aspect of almost all current quarantine strategies is isolation not only of those who test positive for the virus but tracing and isolation of their close contacts as well. The effect of this is neglected here. Can the authors comment on how this might affect their conclusions?

R1 notes that contact tracing is not directly included within our model and asks us to comment on what effect adding it would have to the conclusion. Although we accept that contact tracing is an important factor in controlling the spread of the disease, it is not something that we have directly included in our model. A good contact tracing strategy is something that would increase the prevalence of the tested population in Figure 7, but studying the issue would require significant changes to the model due to the fact that isolating an individual for a period of time requires knowing which of those have been isolated as a result of proximity to an infected individual. This could be modelled without an agent based model, but would require a separate state in the system to represent ‘susceptible-isolated’. This would be a more sophisticated model, and would indeed be an interesting additional factor to consider. But our model is intended to demonstrate the impact of diagnostic uncertainty, and adding this degree of complexity at this point is unlikely to impact the findings that diagnostic uncertainty has a significant effect on disease dynamics. We have added additional text to the paper to explain this. 

Reviewer #1 - Comment 4. A large amount of real world data is now available about COVID19. Can the authors test their analysis and conclusions using any of these data sets? The robustness of their conclusions can be significantly improved if even a partial comparison is presented.

R1 also asks whether the model could be updated to be compared to real-world data sets, this is something we explored when developing the model but decided was not required. We made this decision as the model is not intended to be a true reflection of the dynamics of the COVID-19 epidemic. The model is intended to demonstrate the impact of diagnostic uncertainty on the spread of an epidemic and show that such effects are non-negligible, which we feel it achieves. Additionally, any calibration of the model that was performed on the data that was available in April when the paper was written is likely to be out of date by now (August) and any calibration performed now would be out of date at the time of the paper’s publication. Neither of these calibrations would affect the conclusions of the paper. We have added text when introducing the model as well as in the conclusions section to explain this point.

Reviewer #1 - Comment 5. In lines 280-281, a 10-fold lower viral testing capacity (10,000/day) is used compared to that (100,000/day) used earlier in the article without providing a justification. As of now, it seems the 100,000 number has in fact been surpassed in the UK. Does this change conclusions?

Test capacities have been aligned across the different scenarios to aid comparison. The impact of increasing the test capacity on the affected figures is negligible, though naturally a higher rate of testing affects the dynamics quite significantly. The general conclusion from Figure 6 was that sensitivity doesn’t affect the peak infection count for the immunity passports scenario, though it did allow for a more rapid rate of release from quarantine, and this holds with the updated figure though now a second peak is evident in all cases of low prevalence (Prev<0.01). Figure 7 changes more significantly, as now a second peak is evident in more cases. But again, the general conclusion that a low specificity test for ‘immunity passports’ has the potential to exacerbate peak infections unless prevalence is already extraordinarily high still holds, which an increased test capacity only makes more pronounced. The figure still supports great caution in using antibody testing to justify easing lockdown measures.

Reviewer #1 - Comment 6. An assumed UK context is implicitly used at a number of places in the article. For clarity, these should be explicitly stated – including all numbers of population or parameters used that depend on this context.

We believe that the manuscript holds more potential when considered beyond the scope of the UK, and we do agree that there is a strong implicit UK context throughout. We have attempted to relieve this somewhat, and have made assumptions of populations and prevalence etc. more explicit. We have included tables of parameters for each analysis case for reproducibility.

Reviewer #1 - Minor Comment 1. The amount of detail in which textbook definitions of diagnostic quality parameters is reproduced here – while potentially useful to the lay reader – is not necessary. Citations to relevant texts or other sources can suffice.

R1 takes issue with the amount of review material in the paper, something about which we are acutely self-conscious. But readers of the preprint have praised the fact that PPV and NPV are so clearly explained, and we believe this is essential to motivate the implications of the uncertainty about the model parameters. For instance, a journal club at Manchester University including Paul Klapper, Professor of Clinical Virology, strongly lauded the clarity of re-stating the definitions of these terms which are so important to the intent of the manuscript.(https://youtu.be/IvHYzuKZFRs?t=1819 ) We feel this is a reasonable justification to retain the explanation of these terms which amounts to less than a page of the text. 

Reviewer #1 - Minor Comment 2a. Similarly, Figure 1 and 2 – to the extent that they are needed at all – would benefit from being converted to a plot showing variation of PPV with prevalence instead.

Finally, R1 believes that Figures 1 and 2 would benefit from being converted to a plot showing variation of PPV with prevalence instead. Figures 1 & 2 mirror the pedagogical approach of Gigerenzer in presenting Bayes’ rule in terms of natural frequencies which are a highly effective means of conveying conditional probabilities (https://doi.org/10.1136/bmj.d6386). A graph could not do this. Again, we appreciate that there is a lot of review information in the manuscript, but we believe this significantly improves its readability and interpretability of our findings. 

Reviewer #1 - Minor Comment 2b. In general, in this reviewer’s opinion, the authors here adopt a more journalistic or colloquial tone in their writing than is usual in scientific literature. A significant fraction of the citations are from popular media as well. All of this only ends up distracting the reader from the scientific content. This is avoidable and can be easily rectified.

We agree that the tone of the paper takes a more colloquial tone, and that popular media citations were frequent and distracting. We have taken pains to remove all of these citations other than those which we felt were important for the context of the manuscript. Tonally we feel that the manuscript achieves the desired purpose, and again we may point to the digestibility noted by the Manchester University journal club as supportive of the approach taken.

Reviewer #2 - Comment 1: The availability of the model implementation codes (which platform/language?) would improve the impact by enabling the testing of new strategies for the relaxation of current social distancing measures outside the 3 tested scenarios by the readers. Additionally, use of tables to fully report the parameters that are kept constant in each scenario/plot are required for reproducibility of the results.

R2 also asks us to make the model available to readers of the paper and to show what parameters were used to generate the figures in the paper. We have made the implemented code available via Github (https://github.com/Institute-for-Risk-and-Uncertainty/SIRQ-imperfect-testing) and introduced tables describing the parameter values employed.

Reviewer #2 - Comment 2: While the temporal plots and the choice of parameters were selected wisely to emphasize the key points of the paper and interesting dynamics, the complex interactions of the key variables of the model call for more rigorous analysis to fully capture the nonlinear dynamics. As hinted in the top left two panels of Figure 6, nonlinear dynamics are observed such as oscillatory and dampening dynamics. These call for additional rigorous analysis such as sensitivity analysis to key parameters of the system in each scenario, or parameter sweeps, phase portraits, or depicting the phenotypic spaces (e.g. key dynamical behaviors in the two-dimensional space of p and tau_B).

Reviewer 2, Sepideh Dolatshahi (R2), thought the model in the paper was “simple yet powerful”, and she asks for more analysis to be performed on the model to fully capture the non-linear dynamics shown as well as a sensitivity analysis to explore the relationships between key parameters. This is a very interesting point, and analysis of this aspect of epidemiological dynamics would certainly be a very fruitful area of research when considering future strategies and modelling. However, the intent of this paper was to demonstrate that diagnostic uncertainty can have a significant impact on disease dynamics when used to justify quarantine and release strategies. Although rigorous analysis of the dynamics of this model is not within the scope of the current paper, we have added text in the conclusions mentioning the need for dynamical analysis and sensitivity analysis in future modelling.

Reviewer #2 - Minor Comment 1: Typo in text line 128, PPV=0.95, 0.8 is not correct. (as also noted in Figure caption).

This has been resolved, the figures were indeed incorrect.

Reviewer #2 - Minor Comment 2: Although expected from the definition of Bayes formula, I think it would be beneficial to emphasize from the beginning that prevalence means different things in the viral and antibody tests.

We agree that this is the cause of some confusion. We have altered the terminology to use ‘seroprevalence’ when referring to the presence of antibodies in the population.

We thank the reviewers for their thoughtful comments. We feel the changes made to respond to their suggestions have significantly improved the manuscript, which we are pleased to resubmit for your consideration. 

Kind regards

Nicholas Gray, et al.

---

## [Decision Letter · Decision Letter 1]

5 Oct 2020

Is "No test is better than a bad test''? Impact of diagnostic uncertainty in mass testing on the spread of COVID-19

PONE-D-20-13370R1

Dear Dr. Gray,

We’re pleased to inform you that your manuscript has been judged scientifically suitable for publication and will be formally accepted for publication once it meets all outstanding technical requirements.

Kind regards,

Jishnu Das, Ph.D.

Academic Editor

PLOS ONE

Additional Editor Comments (optional):

Please fix the typo noted by Reviewer 2. The figures are fine as is - Figs 1/2 and 3/4 do not need to be combined.

Reviewers' comments:

Reviewer's Responses to Questions

**Comments to the Author**

1. If the authors have adequately addressed your comments raised in a previous round of review and you feel that this manuscript is now acceptable for publication, you may indicate that here to bypass the “Comments to the Author” section, enter your conflict of interest statement in the “Confidential to Editor” section, and submit your "Accept" recommendation.

Reviewer #1: All comments have been addressed

Reviewer #2: (No Response)

2. Is the manuscript technically sound, and do the data support the conclusions?

Reviewer #1: Yes

Reviewer #2: Yes

3. Has the statistical analysis been performed appropriately and rigorously? 

Reviewer #1: Yes

Reviewer #2: N/A

4. Have the authors made all data underlying the findings in their manuscript fully available?

Reviewer #1: Yes

Reviewer #2: Yes

5. Is the manuscript presented in an intelligible fashion and written in standard English?

Reviewer #1: Yes

Reviewer #2: Yes

6. Review Comments to the Author

Reviewer #1: (No Response)

Reviewer #2: Previous minor Comment 1 (Typo in text line 128, PPV=0.95, and not 0.8) was acknowledged, although the text was incorrect not the figure. However, it was not fixed (now line 110).

Other than this, the authors have addressed my comments.

Minor comment: Maybe this is an editorial decision, but in my opinion the flow of the manuscript can benefit from combining Figures 1 and 2 (New Fig. 1 A,B) and combining Figures 3 and 4 (New Fig. 2A,B).

7. PLOS authors have the option to publish the peer review history of their article (what does this mean?). If published, this will include your full peer review and any attached files.

Reviewer #1: No

Reviewer #2: **Yes: **Sepideh Dolatshahi

---

## [Editor Report · Acceptance letter]

12 Oct 2020

PONE-D-20-13370R1 

Is “no test is better than a bad test”? Impact of diagnostic uncertainty in mass testing on the spread of COVID-19 

Dear Dr. Gray:

I'm pleased to inform you that your manuscript has been deemed suitable for publication in PLOS ONE. Congratulations! Your manuscript is now with our production department. 

Kind regards, 

on behalf of

Dr. Jishnu Das 

Academic Editor

PLOS ONE